# The diagnostic performance of combined conventional cytology with smears and cell block preparation obtained from endoscopic ultrasound-guided fine needle aspiration for intra-abdominal mass lesions

**Nonthalee Pausawasdi**[1,2]*, **Penprapai Hongsrisuwan**[1], **Wipapat Vicki Chalermwai**[3], **Amna Subhan Butt**[1,4], **Kotchakon Maipang**[1], **Phunchai Charatchareonwitthaya**[1,2]

1 Siriraj GI Endoscopy Center, Faculty of Medicine Siriraj Hospital, Mahidol University, Bangkok, Thailand, 2 Division of Gastroenterology, Department of Medicine, Faculty of Medicine Siriraj Hospital, Mahidol University, Bangkok, Thailand, 3 Department of Pathology, Faculty of Medicine Siriraj Hospital, Mahidol University, Bangkok, Thailand, 4 Section of Gastroenterology, Department of Medicine, Aga Khan University Hospital, Karachi, Pakistan

* nonthaleep7@gmail.com

## Abstract

### Background/Aim

Endoscopic ultrasound-guided fine needle aspiration (EUS-FNA) is the primary method for tissue acquisition of intra-abdominal masses. However, the main limitation of cytology alone is the lack of tissue architecture and inadequate samples. This study aimed to evaluate the diagnostic performance of combined conventional cytology and cell block preparation obtained from EUS-FNA of intra-abdominal masses without Rapid On-site Evaluation (ROSE).

### Methods

Cytologic smears and cell block slides of 166 patients undergoing EUS-FNA during 2010–2015 were reviewed by an experienced cytopathologist blinded to clinical data.

### Results

125 patients had neoplastic lesions. Pancreatic adenocarcinoma was the most common etiology (35.5%), followed by lymph node metastasis (27.7%). The mean mass size was 2.5 ±1.3 cm. The mean number of passes was 1.9±1.28. Tissue adequacy for conventional cytology and cell block preparation was 78.9% and 78.1%, respectively. Factors associated with tissue adequacy were assessed. For cytology, lesions of > 2.1 cm, masses in the pancreatic body or tail, malignancy, and pancreatic cancer were positively associated with adequate cellularity. For cell block preparation, lesions of > 3 cm and malignancy were associated with increased tissue adequacy. The conventional cytology alone had a sensitivity of 68.5%, a specificity of 95.7%, and an area under the receiver operating characteristics

**Data Availability Statement:** All relevant data are within the paper and its Supporting Information files.

**Funding:** The authors received no specific funding for this work.

**Competing interests:** The authors have declared that no competing interests exist.

**Abbreviations:** AFP, alpha-fetoprotein; AUROC, Area under the receiver operating characteristics; CAM, cell adhesion molecule; CD, cluster of differentiation; CDX-2, caudal-type homeobox transcription factor 2; CI, Confidence interval; CK, cytokeratin; DOG-1, discovered on gastrointestinal stromal tumors; EUS, Endoscopic ultrasound; EUS-FNA, Endoscopic ultrasound-guided fine needle aspiration; ESGE, European Society of Gastrointestinal Endoscopy; FNB, Fine needle biopsy; FFPE, Formalin-fixed paraffin-embedded; GI, Gastrointestinal; GIST, Gastrointestinal stromal tumor; Hep par 1, hepatocyte paraffin 1 monoclonal antibody; IPMN, intraductal papillary mucinous neoplasm; IQR, interquartile range; MOSE, macroscopic on-site evaluation; NPV, Negative predictive value; OD, Odds ratio; PAP, Papanicolaou; Pax, paired box gene; PPV, Positive predictive value; RCC, renal cell carcinoma; ROSE, Rapid On-site Evaluation; SD, Standard deviation; S-100, S-100 protein; TTF-1, thyroid transcription factor 1 (TTF-1).

(AUROC) of 0.821. The cell block preparation alone had a sensitivity of 65.4%, a specificity of 96%, and an AUROC of 0.807. The combined conventional cytology and cell block preparation performed significantly better than either method alone (p<0.05), as demonstrated by an increased AUROC of 0.853. Furthermore, cell block detected malignancy in 9.3% of cases where the cytologic smears failed to identify malignant cells.

## Conclusions

The combined conventional cytology and cell block preparation increased the diagnostic accuracy of EUS-FNA compared to either method alone. This approach should be implemented in routine practice, especially where ROSE is unavailable.

## Introduction

The management of patients with intra-abdominal mass lesions can be challenging in clinical practice as cross-sectional imaging alone is not always sufficient to provide the diagnosis, therefore tissue sampling may be required to decide optimal therapeutic options. Traditionally, ultrasound and computed tomography-guided biopsies were used to obtain tissue samples from such lesions [1]. Over the years, endoscopic ultrasound (EUS)-guided fine needle aspiration (EUS-FNA) has evolved and become the primary technique for tissue acquisition of intra-abdominal lesions adjacent to the gastrointestinal tract, including subepithelial masses arising from the gut wall, pancreatic lesions, lymph nodes, mesentery, and masses in other solid organs [2–4]. This technique is preferable compared to percutaneous and surgical guided biopsies due to lower morbidity and mortality, possibly lower risk of tumor seeding, and cost-effectiveness [5].

The diagnostic accuracy of EUS-FNA ranges from 64% to 96% [4, 6–8]. This variability in the diagnostic yield could be attributed to the location of the tissue, level of expertise, and the presence of rapid on-site evaluation (ROSE). Several studies have demonstrated that ROSE improves the tissue adequacy and diagnostic accuracy of EUS-FNA; however, this facility is not always available [9–12]. The need for a cytotechnician or cytopathologist to be present in the room during the endoscopic procedure, the lack of dedicated cytopathologists, the increased workload for cytopathologists, and the cost limit the use of ROSE in many areas. Despite being the primary technique for tissue acquisition of gastrointestinal and pancreatic diseases, the lack of tissue architecture and inadequate samples are the main limitations of this method, thus, diagnosing certain conditions such as lymphoma, neuroendocrine tumor, auto-immune pancreatitis, and gastrointestinal stromal tumor (GIST) can be challenging because tissue architecture and additional immunostaining are often required [2, 13].

Several factors, particularly the newly developed biopsy needles and tissue handling and processing techniques, have been investigated to overcome these limitations. The initial biopsy needle design known as a tru-cut needle is technically challenging and increases the risk of complications [14–16]. In contrast, the recently developed fine needle biopsy (FNB) with a side port or specially designed needle tip for core tissue collection has been shown to improve tissue adequacy for histological analysis and obviate the need for ROSE [17–19]. Nonetheless, the use of FNB may be limited in resource-limited areas due to increased expenses. Cell block preparation obtained during conventional EUS-FNA has been increasingly recommended because it allows histological and immunohistochemical examination, especially when ROSE is not available [20, 21].

The current study was conducted with the primary aim to assess the diagnostic performance of combined conventional cytology with smears and cell block preparation obtained from EUS-FNA in intra-abdominal solid masses and compare it to each method alone.

## Materials and methods

### Patient population

A retrospective review of the EUS database at a tertiary care center from 2010 to 2015 was conducted. This study was carried out following the Declaration of Helsinki. The study was approved by the Faculty of Medicine Siriraj Hospital Institutional Review Board (protocol number 111/2557). The informed written consent was waived given the retrospective nature of the study. The patients who underwent EUS-FNA for intra-abdominal masses were identified. The inclusion criteria included 1) age > 18 years, 2) intra-abdominal solid masses accessible by EUS, 3) complete medical records with at least 12 months follow-up, 4) available cytologic smears and cell block slides for review. Exclusion criteria included 1) cystic lesions 2) incomplete medical records and cytopathological data. Patient demographics, clinical presentation, EUS findings, FNA results, complications, and clinical courses were reviewed and analyzed.

### EUS-FNA technique

EUS was performed by an experienced endoscopist, who performed more than 2000 EUS cases at a tertiary care center, using a linear array echoendoscope (GF-UC140P-AL5/AL10, Olympus Corp., Tokyo, Japan). Fine needle aspiration was done using a 22-gauge needle (EZshot; Olympus Medical, Tokyo, Japan, and Echotip; Cook Medical, IN, U.S.A.). The technique involved localization of the target lesion, doppler evaluation, needle puncture, tissue aspiration, and specimen collection. Once the lesion was identified, a doppler ultrasound was used to evaluate the intervening vessels. After identifying the proper window without intervening vessels in the needle passage, the stylet was slightly withdrawn to sharpen the needle tip, and the lesion was punctured under real-time ultrasound guidance. After puncturing, the stylet was removed, and negative pressure was applied with 5 mL of suction, followed by moving the needle to and fro within the lesion using the fanning technique. The suction was stopped before removing the needle from the lesion. The tissue was retrieved, and the stylet was inserted into the needle until the specimens extruded through the needle tip and placed onto glass slides for visual inspection. A macroscopic examination was then performed to identify a few visible whitish core tissues of any length. If inadequate macroscopic material was observed, repeat pass attempts were performed until visible core tissue fragments were obtained. The specimens from each pass were prepared for both cytologic smears and cell block preparation for histological evaluation. The maximum number of passes was six based on a study by Jhala et al. The authors demonstrated that 90% of adequate samples were obtained within 6 passes, after which there was only a slight increase in obtaining an adequate sample [22].

### Cytologic smears and cell block preparation

ROSE was not available in routine practice at our center due to the limited number of dedicated cytopathologists specializing in gastrointestinal and pancreatic diseases. After each pass, the stylet was reinserted into the needle to expel the aspirated material on glass slides. The specimens were smeared onto the slide and immediately fixed with 95% ethyl alcohol solution. The alcohol-fixed slides were stained with Papanicolaou stain for cytological examination. The remaining material was placed into a bottle containing 10% formalin solution for cell block

preparation. Formalin-fixed tissues were embedded in paraffin, and then slides were made using the standard technique [23]. The slides were stained with hematoxylin and eosin and reviewed. Depending on morphology and the cytopathologist's decision, additional immuno-histochemical studies were performed on the tissue blocks derived from formalin-fixed paraffin-embedded (FFPE) to diagnose lymphoma, GIST, neuroendocrine tumors, or metastatic cancer of unknown origin. The immunohistochemical stains used to differentiate various carcinomas included AE 1/AE 3, cell adhesion molecule (CAM) 5.2, cluster of differentiation (CD) 45, cytokeratin (CK) 7, CK 20, S-100 protein (S-100), vimentin, desmin, caudal-type homeobox transcription factor 2 (CDX-2), thyroid transcription factor 1 (TTF-1), chromogranin, synaptophysin, hepatocyte paraffin 1 monoclonal antibody (Hep par 1), alpha-fetoprotein (AFP), paired box gene (Pax-8), renal cell carcinoma (RCC), CD 10. The information about antibody clones are provided in S1 Table. To differentiate spindle cell tumors, we used CD 117, discovered on gastrointestinal stromal tumors (DOG-1), CD 34, S100, and desmin. CD 20 was used to diagnose B-cell lymphoma. Other special stains included acid-fast bacilli and congo red. This study retrieved all available cell block prepared from FFPE and 95% alcohol-fixed slides for cytological analysis. Fig 1 demonstrates an EUS image of a pancreatic lesion, specimen handling after aspiration, Papanicolaou smears, and the hematoxylin and eosin stain.

## Definition of cytology and cell block interpretation

All samples were reviewed by an experienced GI cytopathologist blinded to clinical data and prior cytological and cell block results. The tissue adequacy and cytology classification were assessed based on the Papanicolaou society of cytopathology system for reporting pancreaticobiliary cytology [24]. The specimens were considered adequate if the acquired material

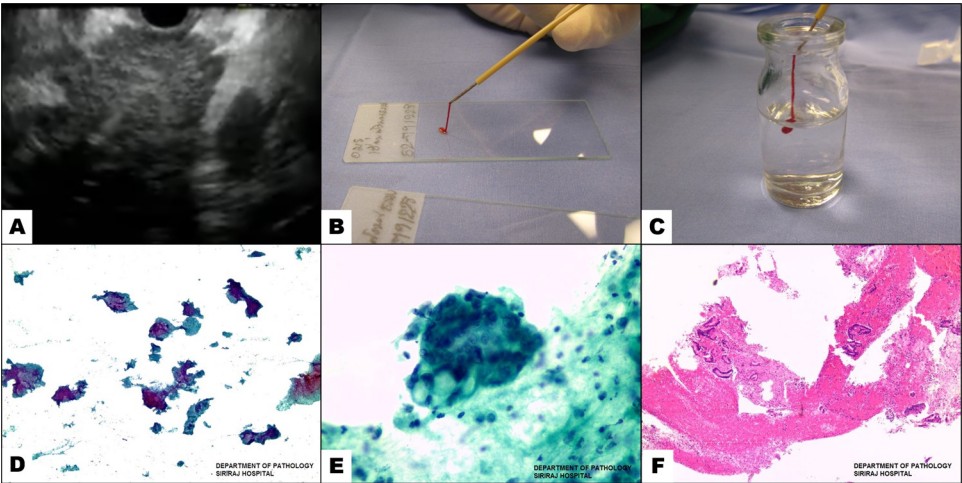

**Fig 1. Endoscopic ultrasound-guided fine needle aspiration with tissue processing method.** (A) Endoscopic ultrasound-guided fine needle aspiration of a pancreatic mass. (B) The specimen was placed onto a glass slide and smeared. (C) The specimen was placed into a bottle containing formalin for cell block preparation. (D) In low power magnification, a cytologic smear shows multiple large fragments of malignant cells in a background of necrotic debris. (Papanicolaou stain, magnification 40X). (E) In high power magnification, a cytologic smear shows a three-dimensional cluster of malignant cells with cellular crowding, nuclear pleomorphism, irregular nuclear membrane, and coarsely clumped chromatin in a background of necrotic debris. (Papanicolaou stain, magnification 400X). (F) Cell block preparation shows tissue fragments containing infiltrating poorly formed glandular structures containing dysplastic cells. The malignant cells are surrounded by desmoplastic stroma and blood clots (Hematoxylin & Eosin stain, magnification 40X).

provided an acceptable amount of cell from the target lesion for cytological evaluation and adequate cells and tissue architecture for histological assessment. The diagnoses from cytologic smears and cell block preparation were categorized as unsatisfactory, negative for malignancy, atypia, suspicious for malignancy, and positive for malignancy. The diagnosis of malignancy was made if the cytopathology reported positive or suspicious for malignancy. The reports of negative for malignancy and atypia were categorized as no malignancy [25, 26]. In cases of inadequate tissue, the EUS-FNA was repeated if clinical presentations or radiological imaging were suspicious for malignancy.

## Diagnostic criteria

The final diagnosis was established by 1) histology from a surgical specimen, 2) cell block interpretation and immunohistochemical stain from FFPE tissue obtained via EUS-FNA, 3) cytological diagnosis, 4) minimum of 12 months follow-up for clinical evaluation and interval imaging for benign lesions. A patient was finally diagnosed with malignancy if there was (1) evidence of malignancy based on cytologic smears or cell block material obtained via (a) EUS-FNA, (b) surgical pathology, or (2) cytopathological results suspicious for malignancy and clinical courses suggesting malignancy. The clinical evidence of malignancy comprised (a) new radiographic abnormalities, including regional or distant metastasis, mass infiltrating blood vessels, or adjacent organs (b) cancer-related mortality. The diagnosis of benign conditions required negative cytopathological assessment and at least 12 months of clinical and imaging follow-up, demonstrating no progression or resolution of the lesions.

## Statistical analysis

The data was analyzed using SPSS version 16 (SPSS, Inc., Chicago, Illinois). Continuous variables were summarized as mean ± standard deviation and categorical variables as percentages. Comparison between groups was performed using the $\chi 2$ test for categorical variables and the t-test for continuous variables. The area under the receiver operating characteristics (AUROC) curve was constructed to evaluate the overall accuracy of cytology and cell block preparation and compared techniques with the DeLong test. The sensitivity, specificity, positive predictive value (PPV), and negative predictive value (NPV) with 95% confidence intervals (95% CI) were determined for each method. We also made the evaluation based on the different possible outcomes indicative of malignancy for each method. Logistic regression analysis was used to determine factors affecting tissue adequacy. The data were presented as an odds ratio (OD) with a 95% CI. A p-value of less than 0.05 was considered statistical significance.

## Results

### Characteristics of the study population

Three hundred and thirty-five patients underwent EUS-FNA for intra-abdominal mass lesions during the study period. One hundred and forty-five patients were excluded due to cystic lesions. Specimens were not available for review in 24 cases; thus, 166 patients were included in the study. The mean age was 58 ± 14 years (18 to 84 years). Abdominal pain was the most common presentation accounting for 34.9%, followed by weight loss (23.4%), jaundice (12.7%), and abdominal masses (4.8%). Cross-sectional imaging demonstrated intra-abdominal masses in 141 patients (84.9%). Of the 166 patients, 125 (75.3%) had neoplastic lesions, and 41 (24.7%) had benign conditions. The final diagnoses included pancreatic adenocarcinoma (35.5%), metastatic lymph nodes (27.7%), inflammatory/reactive change (24.7%), lymphoma (3.6%), gastrointestinal stromal tumor (3.6%), cholangiocarcinoma (1.8%), malignant

intraductal papillary mucinous neoplasms (1.2%), sarcoma (0.6%), neuroendocrine tumor (0.6%), and hepatocellular carcinoma (0.6%), as shown in Table 1.

## Final diagnosis

Among patients with neoplastic lesions, surgical pathology was available to confirm the diagnoses in 16 patients (12.8%). Surgical pathology revealed pancreatic adenocarcinoma (n = 8), gastrointestinal stromal tumor (n = 3), malignant intraductal papillary mucinous neoplasm (n = 1), distal cholangiocarcinoma (n = 1), lymphoma (n = 1), gastric cancer (n = 1) and metastatic carcinoma (n = 1). EUS-FNA guided cytohistological analysis demonstrated malignancies in 76 patients. Both cytologic smears and cell block preparation detected malignancies in 59 patients, cytologic smears alone in 9 patients, and cell block preparation alone in 8 patients. The final diagnoses based on surgical pathology and EUS-FNA guided cytohistological analysis and clinical courses are shown in Table 2. The specimens from cell block preparation obtained from FFPE allowed ancillary studies, including immunohistochemical and acid-fast bacilli stains in 31 cases (18.7%), providing the diagnosis of tuberculous lymphadenitis (n = 12), lymphoma (n = 5), pancreatic adenocarcinoma (n = 4), mesenchymal tumor (n = 4), neuroendocrine tumor (n = 4), hepatocellular carcinoma (n = 1), and IgG4 associated pancreatitis (n = 1).

## EUS findings

The mean mass size was 2.5 ± 1.3 cm (range 0.7 to 10 cm); approximately 40% of lesions were smaller than 2 cm, 32% were 2–3 cm, and 28% were larger than 3 cm. The most common site

**Table 1. Patient characteristics and the definite diagnosis.**

| Parameters | Value |
|---|---|
| Age, year (mean ± SD) | 58 ± 14 |
| Sex, n (%) | |
| Male | 90 (54.2) |
| Female | 76 (45.8) |
| Clinical manifestation, n (%) | |
| Abdominal pain | 58 (34.9) |
| Weight loss | 39 (23.4) |
| Jaundice | 21 (12.7) |
| Abdominal mass | 8 (4.8) |
| Abnormal imaging | 141 (84.9) |
| Definite diagnosis, n (%) | |
| Pancreatic adenocarcinoma | 59 (35.5) |
| Metastatic lymph nodes | 46 (27.7) |
| Inflammatory/Reactive change | 41 (24.7) |
| Lymphoma | 6 (3.6) |
| Gastrointestinal stromal tumor | 6 (3.6) |
| Cholangiocarcinoma | 3 (1.8) |
| Malignant IPMN | 2 (1.2) |
| Sarcoma | 1 (0.6) |
| Neuroendocrine tumor | 1 (0.6) |
| Hepatocellular carcinoma | 1 (0.6) |

NOTE. Data are presented as mean ± standard deviation or the number (%) of patients with a condition.

SD, standard deviation; IPMN, intraductal papillary mucinous neoplasms.

**Table 2. Final diagnoses based on surgical pathology and cytohistological analysis via EUS-FNA.**

| Final diagnoses | Surgical pathology | EUS-FNA with cytohistological analysis and clinical courses |
|---|---|---|
| Malignancy, n (%) | 16 (9.6) | 109 (65.7) |
| Pancreatic adenocarcinoma | 8 (4.8) | 51 (30.7) |
| Metastatic carcinoma | 1 (0.6) | 45 (27.1) |
| Lymphoma | 1 (0.6) | 5 (3.0) |
| Gastrointestinal stromal tumor | 3 (1.8) | 3 (1.8) |
| Distal cholangiocarcinoma | 1 (0.6) | 2 (1.2) |
| Gastric cancer | 1 (0.6) | 1 (0.6) |
| Malignant IPMN | 1 (0.6) | 1 (0.6) |
| Pancreatic neuroendocrine tumor | n/a | 1 (0.6) |
| Benign, n (%) | 0 | 41 (24.7) |

NOTE. Data are presented as the number (%) of patients with a condition. IPMN, Intraductal Papillary Mucinous Neoplasm.

was the pancreas (51.8%), followed by intra-abdominal lymph nodes (37.9%) and bowel wall (6%). Approximately 80% of the lesions were hypoechoic suggestive of solid lesions. The mean number of needle passes was 1.9±1.3 (range, 1–6). The endosonographic characteristics of the lesions are summarized in Table 3. There were no adverse events related to the EUS-FNA.

## Tissue adequacy

The percentage of tissue adequacy for cytology and cell block preparation was 78.9% and 78.1%, respectively. Of the 35 patients who had inadequate cellularity for cytologic smears, 6 patients (17.1%) had adequate specimens for cell block preparation. Factors associated with tissue adequacy were location, size, and the nature of lesions, as shown in Table 4. For cytology, lesions of > 2.1–3 cm, masses located in the pancreatic body and tail, malignancy, and pancreatic cancer were positively associated with adequate cellularity. In contrast, intra-abdominal

**Table 3. Endosonographic characteristics of intra-abdominal mass lesions.**

| Characteristics | Value |
|---|---|
| Size, cm | 2.5 ± 1.3 |
| Number of needle passes | 1.9 ± 1.3 |
| Echogenicity of the lesion, n (%) | |
| Hypo-echoic | 131 (78.9) |
| Hetero-echoic/mixed | 32 (19.3) |
| Iso-echoic | 2 (1.2) |
| Hyper-echoic | 1 (0.6) |
| Location of the lesion, n (%) | |
| Pancreas | 86 (51.8) |
| • Head | 49 (56.9) |
| • Body | 26 (30.2) |
| • Tail | 11 (12.7) |
| Intra-abdominal lymph node | 63 (37.9) |
| Bowel wall | 10 (6.0) |
| Liver | 4 (2.4) |
| Retroperitoneal mass | 3 (1.8) |

NOTE. Data are presented as mean ± standard deviation or the number (%) of patients with a condition.

**Table 4. Factors associated with tissue adequacy of specimens obtained by EUS-FNA.**

| Factor | Cytology | | | | Cell block preparation | | | |
|---|---|---|---|---|---|---|---|---|
| | Adequacy (n = 131) | Inadequacy (n = 35) | Unadjusted OD (95% CI) | P-value | Adequacy (n = 125) | Inadequacy (n = 41) | Unadjusted OD (95% CI) | P-value |
| Number of needle passes, median (IQR) | 2 (1–3) | 1 (1–3) | 1.27 (0.94–1.72) | 0.116 | 1 (1–3) | 1 (1–3) | 1.11 (0.84–1.46) | 0.477 |
| Pancreatic lesion, n (%) | | | | | | | | |
| Head | 39 (29.8) | 10 (28.6) | 1.06 (0.47–2.41) | 0.890 | 37 (29.6) | 12 (29.3) | 1.02 (0.47–2.20) | 0.968 |
| Body/Tail | 36 (27.5) | 1 (2.9) | 12.9 (1.70–97.6) | 0.013 | 31 (24.8) | 6 (14.6) | 1.92 (0.74–5.01) | 0.180 |
| Intra-abdominal lymph nodes, n (%) | 42 (32.1) | 21 (60.0) | 0.31 (0.15–0.68) | 0.003 | 44 (35.2) | 19 (46.3) | 0.63 (0.31–1.29) | 0.204 |
| Size of the lesion, n (%) | | | | | | | | |
| <2 cm | 25 (19.1) | 17 (48.6) | 1 (Reference) | | 25 (20.0) | 17 (41.5) | 1 (Reference) | |
| 2.1–3.0 cm | 38 (29.0) | 11 (31.4) | 2.55 (1.02–6.40) | 0.045 | 38 (30.4) | 11 (26.8) | 2.35 (0.94–5.84) | 0.066 |
| >3.0 cm | 68 (51.9) | 7 (20.0) | 7.18 (2.64–19.5) | <0.001 | 62 (49.6) | 13 (31.7) | 3.24 (1.37–7.65) | 0.007 |
| Echogenicity of the lesion, n (%) | | | | | | | | |
| Hypo-echoic | 103 (78.6) | 28 (80.0) | 0.58 (0.20–1.63) | 0.299 | 97 (77.6) | 34 (82.9) | 0.71 (0.29–1.78) | 0.470 |
| Hetero-echoic | 27 (20.6) | 5 (14.3) | 1.56 (0.55–4.39) | 0.402 | 26 (20.8) | 6 (14.6) | 1.53 (0.58–4.03) | 0.388 |
| Final diagnosis, n (%) | | | | | | | | |
| Malignancy | 108 (82.4) | 17 (48.6) | 4.97 (2.23–11.1) | <0.001 | 100 (80.0) | 25 (61.0) | 2.56 (1.19–5.50) | 0.016 |
| Pancreatic cancer | 57 (43.5) | 4 (11.4) | 10.8 (3.28–35.3) | <0.001 | 51 (40.8) | 10 (24.4) | 2.14 (0.96–4.74) | 0.062 |
| Lymphoma | 5 (3.8) | 1 (2.9) | 3.91 (0.42–36.5) | 0.231 | 5 (4.0) | 1 (2.4) | 1.67 (0.19–14.7) | 0.646 |
| Inflammatory disease | 23 (17.6) | 18 (51.4) | 0.20 (0.09–0.45) | <0.001 | 25 (20.0) | 16 (39.0) | 0.39 (0.18–0.84) | 0.016 |

CI, confidence interval; IQR, interquartile range; OD, odds ratio.

lymph nodes and inflammatory lesions (e.g., chronic pancreatitis and tuberculosis) were negative predictors for tissue adequacy. For cell block preparation, lesions > 3 cm and malignancy were associated with increased tissue adequacy, whereas inflammatory lesions were associated with decreased tissue adequacy.

## Diagnostic performance of cytology and cell block preparation

The performance of cytology and cell block preparation for diagnosing malignancy are shown in Table 5. Overall, cytology had an intermediate sensitivity of 53.7% but a high specificity of 95.7% for diagnosing cancer when only cytological diagnosis classified as positive for malignancy was considered positive. It is noteworthy that cytological analysis categorized as suspicious for malignancy had a specificity and PPV of 100%; therefore, it should be considered equivalent to that classified as positive for malignancy. If both suspicious and positive for malignancy were considered positive, the sensitivity increased to 68.5%, and the specificity remained at 95.7%. Similarly, cell block preparation had an intermediate sensitivity of 57.4% and a high specificity of 95.7% for diagnosing malignancy when only positive for malignancy was considered positive. When only suspicious results were analyzed, the sensitivity was as low as 7.92%, whereas the specificity was 100%. The sensitivity of cell block preparation increased to 65.4% when both positive and suspicious results were considered positive. However, the diagnostic performance of cell block preparation and cytology for diagnosing malignancy were comparable (the AUROCs were 0.807 vs. 0.821, respectively; $p = 0.618$). For combined cytology and cell block preparation, when both positive and suspicious results of either test were considered positive, the sensitivity increased significantly from 64.2% to 74.6%, and the

**Table 5. Diagnostic performance of cytology and cell block preparation obtained by EUS-FNA of intra-abdominal masses.**

| | AUROC (95%CI) | Sensitivity (%) (95%CI) | Specificity (%) (95%CI) | PPV (%) (95%CI) | NPV (%) (95%CI) |
|---|---|---|---|---|---|
| **Cytology** | | | | | |
| Positive or suspicious | 0.821 | 68.5 | 95.7 | 98.7 | 39.3 |
| | (0.744–0.882) | (58.9–77.1) | (78.1–99.1) | (91.6–99.8) | (32.6–46.4) |
| Positive | 0.747 | 53.7 | 95.7 | 98.3 | 30.6 |
| | (0.663–0.819) | (48.3–63.3) | (78.1–99.9) | (89.4–99.7) | (26.1–35.4) |
| Suspicious | 0.574 | 14.8 | 100 | 100 | 20.0 |
| | (0.485–0.660) | (8.7–22.9) | (85.2–100) | | (18.8–21.3) |
| **Cell block preparation** | | | | | |
| Positive or suspicious | 0.807 | 65.4 | 96.0 | 98.5 | 40.7 |
| | (0.727–0.872) | (55.2–74.5) | (79.6–99.9) | (90.6–99.8) | (34.1–47.6) |
| Positive | 0.767 | 57.4 | 96.0 | 98.3 | 35.8 |
| | (0.684–0.838) | (47.2–67.2) | (79.6–99.9) | (89.4–99.7) | (30.5–41.5) |
| Suspicious | 0.540 | 7.92 | 100 | 100 | 21.2 |
| | (0.449–0.629) | (3.5–15.0) | (86.3–100) | | (20.2–22.2) |
| **Cytology and cell block preparation** | | | | | |
| Either test with | 0.853 | 74.6 | 96.0 | 98.8 | 46.2 |
| positive or suspicious | (0.782–0.908) | (65.4–82.4) | (79.6–99.9) | (92.3–99.8) | (38.1–54.4) |
| Either test with | 0.801 | 64.2 | 96.0 | 98.6 | 38.1 |
| positive | (0.723–0.865) | (54.5–73.2) | (79.6–99.9) | (91.1–99.8) | (32.1–44.5) |
| Either test with | 0.586 | 17.3 | 100 | 100 | 21.6 |
| suspicious | (0.498–0.670) | (10.7–25.7) | (86.3–100) | | (20.1–23.0) |

PPV, positive predictive value; NPV, negative predictive value; CI, confidence interval; AUROC, area under the Receiver-Operating Characteristic Curve.

specificity remained high (96%). The AUROC of combined cytology and cell block preparation for diagnosing cancer was 0.853, and this was significantly better than either cytology ($p = 0.002$) or cell block preparation ($p = 0.006$) alone.

Furthermore, we explored whether cell block preparation would help detect malignancy when cytology was inadequate or unsuccessful. Of 86 patients who had either inadequate cytology or cytological diagnosis of negative for malignancy, 58 patients had adequate specimens for cell block preparation. Of these, 8 patients (9.3%) were found to have malignancy, including pancreatic and metastatic cancer. Other diagnoses included tuberculous lymphadenitis (n = 10), reactive lymph nodes (n = 9), abscess (n = 3), GIST (n = 1), benign pancreatic tissue (n = 1). The diagnoses could not be made by cell block evaluation in 26 patients. Among these, malignancy (n = 16), tuberculosis (n = 6), and other benign conditions (n = 4) were discovered during clinical and imaging follow-up.

## Discussion

Despite being a primary modality for tissue acquisition for intra-abdominal solid lesions, EUS-FNA carries certain limitations, such as the inability to obtain core tissue for histological features and inadequate sampling. Cell block preparation has been recognized as a powerful technique for assessing tissue architecture and determining its histological features. Results from retrospective studies suggest that histological evaluation from cell block preparation increases the diagnostic accuracy of EUS-FNA for malignancy [20, 21, 27–29]. The European Society of Gastrointestinal Endoscopy (ESGE), based on low-quality evidence, recommends that EUS-guided tissue sampling should include histologic preparations (e.g., cell blocks) and not be limited to cytology [30]. Therefore, we conducted a comparative analysis of the

diagnostic performance between cytology vs. cell block preparation for histological evaluation vs. combined methods in the absence of ROSE.

The present study showed that cytology alone had a sensitivity of 68.5%, a specificity of 95.7%, and an AUROC of 0.821. Cell block preparation alone had a sensitivity of 65.4%, a specificity of 96%, and an AUROC of 0.807. Cytology and cell block preparation combined performed significantly better than either method alone with an AUROC of 0.853. Also, cell block preparation could detect malignancy in patients who had inadequate specimens for cytology or cytological diagnosis classified as negative for malignancy. These findings emphasize the importance of both cytologic smears and cell block preparation during tissue processing.

In the absence of ROSE, the tissue adequacy for cytological diagnosis may decrease by 10–15% [31]. The reported specimen adequacy ranges from 70%-92% for cytology and 68%-86.5% for cell block preparation for histology using different needle sizes (18G, 22G, 25G) [20, 21, 32]. The present study demonstrated similar results with 78.9% adequacy for cytology and 78.1% for cell block preparation. The factors influencing tissue adequacy have been explored. We found that pancreatic body and tail lesions, tumor size of >2 cm, the diagnosis of malignancy, and pancreatic cancer had a positive association with tissue adequacy, whereas lesions of lymph nodes and inflammatory diseases had a negative association with tissue adequacy for cytological evaluation. For cell block preparation, only tumor size of >30 mm and malignancy were associated with increased tissue adequacy. In contrast, inflammatory diseases were associated with decreased tissue adequacy. It is worth mentioning that pancreatic body and tail lesions were associated with increased tissue adequacy for cytological analysis but pancreatic head lesions were not. This may be related to the technical aspect of EUS-FNA. Lesions in the body and tail are generally easier to puncture because the echoendoscope is in a straight position allowing the needle to puncture through easily compared to the head lesions.

In terms of the number of needle passes, the ESGE recommends 3–4 passes in the absence of ROSE (30). However, the present study demonstrated the mean number of passes of 1.9, which is similar to Moeller et al. In their study, the mean number of passes was 1.88 with the diagnostic adequacy of 98.9%, and the sensitivity of 82.9% [21]. We hypothesize that the mean number of passes in our study is lower than the ESGE recommendation because a macroscopic evaluation was performed after each pass, allowing the endosonographer to stop the tissue aspiration once the fragments of tissue core were observed. Recently, the macroscopic on-site evaluation (MOSE) of the aspirates has been introduced; however, the method has not been standardized. A macroscopic visible core of $\geq$ 4 mm in length is associated with a higher diagnostic yield [33]. A randomized trial comparing the diagnostic yield of MOSE to conventional EUS-FNA without ROSE has shown that both techniques provide a similar yield, but MOSE requires fewer passes than the conventional method [34]. These findings demonstrate the diagnostic value of MOSE, especially in the absence of ROSE. Our study performed an onsite macroscopic examination; nonetheless, the criteria were different from MOSE described in the recent literature. We accepted any length of a few visible cores.

There is a variety of specimen handling and processing methods in clinical practice. We used the specimens obtained from each pass for cytologic smears and cell block preparation. Therefore, no additional passes or new needles were required to obtain tissue for the cell block preparation. Direct smears are often performed, and alcohol or saline is used for liquid-based cytology. Formalin is commonly used for histology preservation [30]. Also, cell block preparation using the sodium alginate method, stained with hematoxylin and eosin, and immunohistochemistry have been described [35]. The refinement and standardization of the tissue processing techniques may enhance the diagnostic accuracy of EUS-FNA and deserve further investigation.

Another essential factor influencing the diagnostic performance of cytology and cell block is the categorization and interpretation of specimens in correlation with the clinical course.

We proposed that cytological and cell block with histological assessment classified as suspicious for malignancy should be considered positive because of its excellent specificity and PPV of 100%. In addition, clinical courses should be taken into consideration when malignancy is suspected on cytological or histological evaluation.

The strengths of this study included 1) large sample size, 2) the results can be applied to small (<2 cm) and large (>3 cm) lesions, 3) a dedicated GI cytopathologist, blinded to clinical data, prospectively reviewed both the cytologic smears and slides from cell block preparation to avoid interpretation bias, and 4) a complete clinical data and long-term follow-up period. However, this was a single-center, single endoscopist-based, single cytopathologist, and a retrospective review. A vast majority of patients did not have surgical pathology. Hence, prospective comparative studies or randomized, controlled trials may be warranted.

## Conclusion

The combination of cytology and cell block preparation improves the diagnostic performance of EUS-FNA in intra-abdominal solid masses compared to either method alone. This approach shows promise and should be routinely implemented in clinical practice, especially where on-site cytopathology is unavailable.

## Supporting information

**S1 Table. The antibody clones for immunohistochemistry staining.**
(DOCX)

**S1 Data.**
(PDF)

## Author Contributions

**Conceptualization:** Nonthalee Pausawasdi.

**Data curation:** Nonthalee Pausawasdi, Penprapai Hongsrisuwan, Wipapat Vicki Chalermwai.

**Formal analysis:** Nonthalee Pausawasdi, Penprapai Hongsrisuwan, Phunchai Charatchareonwitthaya.

**Investigation:** Nonthalee Pausawasdi, Wipapat Vicki Chalermwai.

**Methodology:** Nonthalee Pausawasdi, Phunchai Charatchareonwitthaya.

**Project administration:** Nonthalee Pausawasdi.

**Supervision:** Nonthalee Pausawasdi.

**Visualization:** Nonthalee Pausawasdi.

**Writing – original draft:** Nonthalee Pausawasdi, Penprapai Hongsrisuwan, Wipapat Vicki Chalermwai, Amna Subhan Butt, Kotchakon Maipang.

**Writing – review & editing:** Nonthalee Pausawasdi, Phunchai Charatchareonwitthaya.

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
