## [Decision Letter · Decision Letter 0]

27 Oct 2021

PONE-D-21-31814The diagnostic performance of combined cytology and histology obtained from endoscopic ultrasound-guided fine needle aspiration for intra-abdominal mass lesionsPLOS ONE

Dear Dr. Pausawasdi,

Thank you for submitting your manuscript to PLOS ONE. After careful consideration, we feel that it has merit but does not fully meet PLOS ONE’s publication criteria as it currently stands. Therefore, we invite you to submit a revised version of the manuscript that addresses the points raised during the review process.

We look forward to receiving your revised manuscript.

Kind regards,

Vincenzo L'Imperio

Academic Editor

PLOS ONE

Journal Requirements:

Reviewers' comments:

Reviewer's Responses to Questions

**Comments to the Author**

1. Is the manuscript technically sound, and do the data support the conclusions?

Reviewer #1: Yes

Reviewer #2: Yes

2. Has the statistical analysis been performed appropriately and rigorously? 

Reviewer #1: Yes

Reviewer #2: Yes

3. Have the authors made all data underlying the findings in their manuscript fully available?

Reviewer #1: Yes

Reviewer #2: Yes

4. Is the manuscript presented in an intelligible fashion and written in standard English?

Reviewer #1: Yes

Reviewer #2: Yes

5. Review Comments to the Author

Reviewer #1: The paper carries out a comparative analysis of EUS-FNA cytology and histology diagnostic performances in the context of solid intra-abdominal mass lesions, using a large sample size. The factors that positively and negatively correlate with the adequacy of the samples are identified. The two compared methods did not demonstrate different diagnostic performances, but their combination ensured an AUROC significantly better than either method alone.

Major issues:

1) The whole DISCUSSION (starting from line 250) is quite confusing and needs an overall review.

Minor issues:

1) “Supporting info data” layout has some problems, the last column is moved at the end of the document.

2) Figure 1: the quality is poor and the boxes should be properly framed.

3) An abbreviations section is needed.

4) Lines 56-58: “The management of patients with intra-abdominal mass lesions can be challenging in clinical practice as cross sectional imaging alone is not always sufficient to provide the diagnosis, therefore tissue sampling may be required to decide optimal therapeutic options.”

5) Line 59: tissue samples.

6) Lines 68-70: “The lack of tissue architecture and inadequate samples are the main limitations of this method, thus diagnosing certain conditions… [1]”

7) Line 72: the new line should be used here and not in line 74.

8) Line 75: increases.

9) Patient population: lines 162-164 should be moved here.

10) Line 88: why did your analysis consider only the years 2010-2015?

11) Line 132: from a qualitative point of view, none of the 166 cases had any problems?

12) Line 137: rephrase.

13) Line 141: it should be said also here that the point 4) is referred just to benign lesions.

14) Line 141: “A patient was finally diagnosed…”

15) Lines 169-172: this means that the study of the cytology + histology combination was based just on 69 cases? If so, it should be stated.

16) Lines 172 and 173: gastrointestinal and gastric.

17) Lines 175: delete comma after bracket.

18) Table 1: definite diagnosis sum is 164, are 2 cases missing?

19) Table 2: Hetero-echoic/mixed

20) Table 3: specify the meaning of OD. A visual way to point out if a factor has a positive or a negative correlation should be found.

21) Line 212: “... was considered actually positive.”

22) Line 221-223: it should be said if there is significance for Sn and Sp alone.

23) Lines 241-243: “The European Society of Gastrointestinal Endoscopy (ESGE), based on low-quality evidence, recommends that EUS-guided tissue sampling should include histologic preparations (e.g., cell blocks) and not be limited to cytology [28].”

24) Line 293: a single cytopathologist is another limitation.

Reviewer #2: This research article focuses on the cytological evaluation of intra-abdominal mass lesions. In particular, it is a retrospective study comparing the diagnostic performance of samples obtained through endoscopic ultrasound-guided fine needle aspiration (EUS-FNA) and processed in different ways (conventional cytology with smears versus cell block preparation for histology versus the two methods combined) in a setting that does not allow the use of Rapid On-Site Evaluation (ROSE); it also investigates the different factors that affect the adequacy of the cytological sampling through EUS-FNA.

The study focuses on the evaluation of a sampling method which leads to obtaining cytological samples, which are considered small samples, and highlights their diagnostic strength, especially in case of a combination of conventional cytology with smears and cell block preparation for histology. It is well constructed, the discussion is well articulated, well written and fluid and the strengths and limits of the study are clearly defined.

However, there are some issues that need to be addressed:

MAJOR ISSUES

-The study compares cytological samples obtained through EUS-FNA and processed in two different ways, in particular conventional cytology with smears and cell block preparation for histology. In the title and in the text, however, the word "histology" is often used directly in place of “cell block preparation”. In my opinion, the terminology must be revised, highlighting the fact that your group is focusing on cytological samples, which then, in the case of the cell block preparations, are also able to provide further information (e.g. tissue architecture, possibility of immunohistochemical examination) in comparison to conventional cytological smears.

- It is clear from the beginning that ROSE was not used. Since ROSE is one of those elements which more affects the adequacy of the EUS-FNA, it would be useful, starting from the introduction, to explain the reasons why ROSE is not always available and also later, in the section “Materials and methods”, the reasons why it was not possible to use it in your study.

-Fig. 1: you should specify the magnification of the hematoxylin and eosin picture. If possible, you should also add the conventional cytology counterpart.

-Page 12, lines 254-257: you should provide a possible explanation of the low number of needle passes obtained, in comparison to the European Society of Gastrointestinal Endoscopy (ESGE) recommendations.

-Discussion: you should provide a possible explanation of how the location of the lesion affects the adequacy of the sampling.

MINOR ISSUES

-Page 3, line 59: you should add some references from literature regarding traditional methods used to obtain tissue diagnosis from abdominal mass lesions.

-Page 4, line 106-109: you should clarify the criteria used for the macroscopic evaluation/visual inspection of the specimens obtained from EUS-FNA in your study.

-Page 4, line 109: does the number "six passes" derive from your retrospective experience? Was it based on some literature data?

-Page 5, line 117: you should provide a list of the antibodies used for the additional immunohistochemical studies.

-Page 5, line 137: "Repeat tissue acquisition may be performed as indicated": what does this sentence mean?

-Page 7, lines 168-180: in addition to the data in the text, it would be useful to create a table with the diagnosis (malignant and benign) and where they derive from (smears, cell block preparations, surgical pathology); also you should move the paragraph regarding the immunohistochemistry to the end of the section.

-Page 7, lines 162-180: you should report in the text the same percentages reported in Table 1.

-Page 7, lines 178-180: these criteria were already specified before, in the “Material and methods” section.

-Pages 8-9, table 2: you should add in brackets the explanation of the values for "Size, cm" and "Number of needle passes'', as in table 1 ("mean +/- SD")

-Pages 8-9, table 3: you should explain the abbreviations "OD" and "CI"; what does "Reference" mean regarding the "Size of the lesion"?

---

## [Author Response · Author response to Decision Letter 0]

11 Dec 2021

Dear Editor,

We would like to thank the editor and both reviewers for the constructive criticisms, which have helped us improve our manuscript. As suggested, we have revised the manuscript point-by-point in response to the reviewers’ questions. 

Reviewers' comments:

Reviewer #1: The paper carries out a comparative analysis of EUS-FNA cytology and histology diagnostic performances in the context of solid intra-abdominal mass lesions, using a large sample size. The factors that positively and negatively correlate with the adequacy of the samples are identified. The two compared methods did not demonstrate different diagnostic performances, but their combination ensured an AUROC significantly better than either method alone.

Response: We thank the reviewer for taking the time to review our manuscript and providing helpful input to improve the clarity of our manuscript. We have revised our manuscript to address the reviewer’s comments and provide a point-by-point response to the specific questions. 

Major issues:

1) The whole DISCUSSION (starting from line 250) is quite confusing and needs an overall review.

Response: The entire discussion section has been revised to improve the clarity of the manuscript, as suggested by the reviewer. In brief, the first paragraph stated the role of EUS-FNA in the diagnosis of intra-abdominal lesions and the limitations of the test. The second paragraph presented the study's main results, including the diagnostic performance of cytological analysis alone, histological analysis alone, and combined method. The third paragraph discussed the percentage of tissue adequacy from our study compared to the literature and the factors associated with tissue adequacy. The fourth paragraph touched on the number of needle passes and the usefulness of macroscopic onsite evaluation of specimens, especially in the absence of ROSE. The fifth paragraph described various tissue handling and processing methods and how the specimens were processed in our study. The discussion in the sixth paragraph focused on how to interpret cytohistological specimens based on their category in correlation with clinical courses. Lastly, the strengths and weaknesses of the study were stated in the last paragraph.

Minor issues:

1) “Supporting info data” layout has some problems, the last column is moved at the end of the document.

Response: The PDF file of the “supporting info data” has been reformatted and all columns are shown properly.

2) Figure 1: the quality is poor and the boxes should be properly framed.

Response: We have revised Figure 1 to improve the image quality. The boxes are properly framed. Also, we have added images of cytology smears (PAP smear) to compare with the histology slide (hematoxylin and eosin stain) derived from cell block preparation. 

3) An abbreviations section is needed.

Response: An abbreviations section has been added after the abstract. 

4) Lines 56-58: “The management of patients with intra-abdominal mass lesions can be challenging in clinical practice as cross sectional imaging alone is not always sufficient to provide the diagnosis, therefore tissue sampling may be required to decide optimal therapeutic options.”

Response: The changes have been made as suggested. 

5) Line 59: tissue samples.

Response: “tissue diagnosis” has been changed to “tissue samples” as suggested.

6) Lines 68-70: “The lack of tissue architecture and inadequate samples are the main limitations of this method, thus diagnosing certain conditions… [1]”

Response: The changes have been made as suggested. 

7) Line 72: the new line should be used here and not in line 74.

Response: The change has been made as suggested.

8) Line 75: increases.

Response: The change has been made as suggested.

9) Patient population: lines 162-164 should be moved here.

Response: The sentences from line 162-164 have been moved to “patient population” under Materials and Methods section as suggested.

10) Line 88: why did your analysis consider only the years 2010-2015?

Response: Our center started a formal EUS program in 2009. Initially, FNA needles were used in all cases. The trend started to change after Fine needle biopsy (FNB) was introduced to our center in 2016. A vast majority of patients have been included in several FNB-related projects. Therefore, we elected to recruit patients between 2010-2015.

11) Line 132: from a qualitative point of view, none of the 166 cases had any problems?

Response: Besides overall tissue adequacy being 78%, some minor problems were air artifacts and blood contamination on the cytology smears. However, these issues did not significantly interfere with the interpretation of the specimens. We routinely prepared 4-6 slides for cytology. Air artifacts may have occurred in some slides but not in all. We avoided using suction during tissue aspiration for hypervascular lesions to minimize blood contamination. Also, a doppler was used to detect intervening vessels before puncturing in all lesions, as mentioned in the manuscript. 

12) Line 137: rephrase.

Response: The sentence was rephrased to “In cases of inadequate tissue, the EUS-FNA was repeated if clinical presentations or radiological imaging were suspicious for malignancy.”

13) Line 141: it should be said also here that the point 4) is referred just to benign lesions.

Response: We have clarified that the point 4) is referred benign lesions by adding “for benign lesions” as suggested. 

14) Line 141: “A patient was finally diagnosed…”

Response: The change has been made as suggested.

15) Lines 169-172: this means that the study of the cytology + histology combination was based just on 69 cases? If so, it should be stated.

Response: The combination of cytological and histological analysis was performed in all recruited patients (N=166). However, 69 patients had malignant cells detected by both cytology smears and cell block preparation for histological evaluation. Fourteen patients had malignant cells detected by cytological evaluation alone, and 12 patients had malignant cells detected by histological analysis alone. Surgical pathology was available to confirm the diagnoses in 16 patients. We have clarified this issue by adding the subheading “final diagnosis” with a detailed explanation and a table summarizing the explanation (Table 2) to the Results section. The order of the following tables has been changed accordingly.

16) Lines 172 and 173: gastrointestinal and gastric.

Response: Changes have been made as suggested. 

17) Lines 175: delete comma after bracket.

Response: Comma after bracket was deleted.

18) Table 1: definite diagnosis sum is 164, are 2 cases missing?

Response: Table 1 was revised (N=166). Two missing cases were malignant intraductal papillary mucinous neoplasms (IPMN). 

19) Table 2: Hetero-echoic/mixed

Response: The terminology has been modified (mixed was added) as suggested.

20) Table 3: specify the meaning of OD. A visual way to point out if a factor has a positive or a negative correlation should be found.

Response: To point out whether a factor has a positive or negative correlation with tissue adequacy, we added detailed data demonstrating the probability of each factor to be associated with tissue adequacy. The relative probability of each factor was then presented as the odds ratio. Table 3 (the table's order was changed to 4 after revision) has been revised. 

21) Line 212: “... was considered actually positive.”

Response: The change has been made as suggested. “actually” was added.

22) Line 221-223: it should be said if there is significance for Sn and Sp alone.

Response: When both positive and suspicious results of either test were considered positive, the cytological and histological analysis combined significantly increased the sensitivity from 64.2 to 74.6% (p< 0.05) and the specificity remained unchanged at 96%. The change was made by stating that the sensitivity increased significantly. 

23) Lines 241-243: “The European Society of Gastrointestinal Endoscopy (ESGE), based on low-quality evidence, recommends that EUS-guided tissue sampling should include histologic preparations (e.g., cell blocks) and not be limited to cytology [28].”

Response: Changes have been made as suggested. 

24) Line 293: a single cytopathologist is another limitation.

Response: A single cytopathologist was added to the limitations as suggested. 

Reviewer #2: This research article focuses on the cytological evaluation of intra-abdominal mass lesions. In particular, it is a retrospective study comparing the diagnostic performance of samples obtained through endoscopic ultrasound-guided fine needle aspiration (EUS-FNA) and processed in different ways (conventional cytology with smears versus cell block preparation for histology versus the two methods combined) in a setting that does not allow the use of Rapid On-Site Evaluation (ROSE); it also investigates the different factors that affect the adequacy of the cytological sampling through EUS-FNA.

The study focuses on the evaluation of a sampling method which leads to obtaining cytological samples, which are considered small samples, and highlights their diagnostic strength, especially in case of a combination of conventional cytology with smears and cell block preparation for histology. It is well constructed, the discussion is well articulated, well written and fluid and the strengths and limits of the study are clearly defined.

However, there are some issues that need to be addressed:

Response: We thank the reviewer for taking the time to review our manuscript and providing helpful comments to improve the quality of our manuscript. We have revised our manuscript to address the reviewer’s comments and a point-by-point response to the specific questions are provided. 

MAJOR ISSUES

-The study compares cytological samples obtained through EUS-FNA and processed in two different ways, in particular conventional cytology with smears and cell block preparation for histology. In the title and in the text, however, the word "histology" is often used directly in place of “cell block preparation”. In my opinion, the terminology must be revised, highlighting the fact that your group is focusing on cytological samples, which then, in the case of the cell block preparations, are also able to provide further information (e.g. tissue architecture, possibility of immunohistochemical examination) in comparison to conventional cytological smears.

Response: The reviewer’s comments are well taken. As the reviewer suggested, we have made changes to the terminologies related to cell block preparation and histology throughout the manuscript. The term “histology” was modified to “cell block preparation for histological analysis”. Throughout the article, we were more consistent with the terms/descriptions of tissue processing methods. Also, the manuscript title has been changed to “The diagnostic performance of combined conventional cytology with smears and cell block preparation for histological evaluation obtained from endoscopic ultrasound-guided fine needle aspiration for intra-abdominal mass lesions”.

- It is clear from the beginning that ROSE was not used. Since ROSE is one of those elements which more affects the adequacy of the EUS-FNA, it would be useful, starting from the introduction, to explain the reasons why ROSE is not always available and also later, in the section “Materials and methods”, the reasons why it was not possible to use it in your study.

Response: We have added the following reasons to the introduction to explain why ROSE is not always available. "The need for a cytotechnician or cytopathologist to be present in the room during the endoscopic procedure, the lack of dedicated cytopathologists, the increased workload for cytopathologists, and the cost limit the use of ROSE in many areas." These sentences have been highlighted. Regarding our center, ROSE service could not be offered in routine practice because of the lack of human resources and a limited number of dedicated GI cytopathologists. This has been added to the "cytology and histology preparation" under the materials and methods section.

-Fig. 1: you should specify the magnification of the hematoxylin and eosin picture. If possible, you should also add the conventional cytology counterpart.

Response: Figure 1 has been revised. The magnification of each picture has been added. Images of cytologic smears for PAP stain have been added to compare with the histology slide (H&E stain) derived from cell block preparation. 

-Page 12, lines 254-257: you should provide a possible explanation of the low number of needle passes obtained, in comparison to the European Society of Gastrointestinal Endoscopy (ESGE) recommendations.

Response: We hypothesize that the mean number of passes in our study is lower than the ESGE recommendation because a macroscopic evaluation was performed after each pass, allowing the endosonographer to stop the tissue aspiration once the fragments of tissue core were observed. This possible explanation has been added to the discussion. 

-Discussion: you should provide a possible explanation of how the location of the lesion affects the adequacy of the sampling.

Response: The factors affecting tissue adequacy have been elaborated in more detail, and a possible explanation of how the location of the lesion affects the adequacy of the sampling has been added to the discussion. The following paragraph has been added. “The factors influencing tissue adequacy have been explored. We found that pancreatic body and tail lesions, tumor size of >2 cm, the diagnosis of malignancy, and pancreatic cancer had a positive association with tissue adequacy, whereas lesions of lymph nodes and inflammatory diseases had a negative association with tissue adequacy for cytological evaluation. For histological evaluation, only tumor size of >30 mm and malignancy were associated with increased tissue adequacy. In contrast, inflammatory diseases were associated with decreased tissue adequacy. It is worth mentioning that pancreatic body and tail lesions were associated with increased tissue adequacy for cytological analysis but pancreatic head lesions were not. This may be related to the technical aspect of EUS-FNA. Lesions in the body and tail are generally easier to puncture because the echoendoscope is in a straight position allowing the needle to puncture through easily compared to the head lesions.”

MINOR ISSUES

-Page 3, line 59: you should add some references from literature regarding traditional methods used to obtain tissue diagnosis from abdominal mass lesions.

Response: A reference has been added as suggested. Lipnik AJ, Brown DB. Image-Guided Percutaneous Abdominal Mass Biopsy: Technical and Clinical Considerations. Radiol Clin North Am. 2015;53(5):1049-59.

-Page 4, line 106-109: you should clarify the criteria used for the macroscopic evaluation/visual inspection of the specimens obtained from EUS-FNA in your study.

Response: The criteria used for the macroscopic evaluation have been added. The visual inspection should identify a few visible whitish core tissues of any length.

-Page 4, line 109: does the number "six passes" derive from your retrospective experience? Was it based on some literature data?

Response: The number “six passes” is based on a study by Jhala et al. (Jhala NC, Jhala D, Eltoum I, Vickers SM, Wilcox CM, Chhieng DC, et al. Endoscopic ultrasound-guided fine-needle aspiration biopsy: A powerful tool to obtain samples from small lesions. Cancer cytopathology. 2004;102(4):239-46)

The following explanation has been added. The maximum number of passes was six based on a study by Jhala at el. The authors demonstrated that 90% of adequate samples were obtained within 6 passes, after which there was only a slight increase in obtaining adequate sample (22).

-Page 5, line 117: you should provide a list of the antibodies used for the additional immunohistochemical studies.

Response: The list of antibodies has been added. The immunohistochemical stains used to differentiate various carcinomas included AE 1/3, CAM 5.2, CD 45, CK7, CK 17, CK 20, S-100, vimentin, Desmin, CDX-2, TTF-1, chromogranin, synaptophysin, Hep par 1, AFP, Pax-8, RCC, CD 10, cytokeratin. For spindle cell tumor, we used CD 117, DOG-1, CD 34, S100 protein, and Desmin. CD 20 was used to diagnose B-cell lymphoma. Other special stains included acid-fast bacilli and congo red. 

-Page 5, line 137: "Repeat tissue acquisition may be performed as indicated": what does this sentence mean?

Response: This sentence has been rephrased to “In cases of inadequate tissue, the EUS-FNA was repeated if clinical presentations or radiological imaging were suspicious for malignancy.”

-Page 7, lines 168-180: in addition to the data in the text, it would be useful to create a table with the diagnosis (malignant and benign) and where they derive from (smears, cell block preparations, surgical pathology); also you should move the paragraph regarding the immunohistochemistry to the end of the section.

Response: A new table has been created (table 2) as the reviewer suggested. Also, we added the subheading “final diagnosis” with detailed explanation about how the diagnoses were derived along with the new table2 summarizing the findings. The paragraph regarding the immunohistochemistry is included in this section.

-Page 7, lines 162-180: you should report in the text the same percentages reported in Table 1.

Response: Changes have been made as the reviewer suggested. 

-Page 7, lines 178-180: these criteria were already specified before, in the “Material and methods” section.

Response: These sentences have been removed. 

-Pages 8-9, table 2: you should add in brackets the explanation of the values for "Size, cm" and "Number of needle passes'', as in table 1 ("mean +/- SD")

Response: Changes have been made. 

-Pages 8-9, table 3: you should explain the abbreviations "OD" and "CI"; what does "Reference" mean regarding the "Size of the lesion"?

Response: Explanation of the abbreviations have been added below the table. Regarding “reference”, it indicates that the lesion size of 2 cm was used as the reference (value =1) for comparison. The changes have been made and highlighted.

---

## [Decision Letter · Decision Letter 1]

7 Jan 2022

PONE-D-21-31814R1The diagnostic performance of combined conventional cytology with smears and cell block preparation for histological evaluation obtained from endoscopic ultrasound-guided fine needle aspiration for intra-abdominal mass lesionsPLOS ONE

Dear Dr. Pausawasdi,

Thank you for submitting your manuscript to PLOS ONE. After careful consideration, we feel that it has merit but does not fully meet PLOS ONE’s publication criteria as it currently stands. Therefore, we invite you to submit a revised version of the manuscript that addresses the points raised during the review process.

We look forward to receiving your revised manuscript.

Kind regards,

Vincenzo L'Imperio

Academic Editor

PLOS ONE

Journal Requirements:

**Comments to the Author**

Reviewer #1: The authors have assessed the issues clarifying doubts and appropriately modifying the paper, therefore in my opinion the article can be submitted for publication.

Reviewer #2: I would like to thank the authors for all the answers to my comments and the revision of the manuscript. In my opinion there are still some issues that need to be addressed before publication:

-The word "histology" is still often used directly in place of “cell block preparation”. In my opinion, the terminology must be checked through all the text (considering also the abstract and the tables).

-Line 31 and others: ”combined cytological”: you should change it to “cytology” or “conventional cytology” through all the text.

-Lines 127-130: “Three hundred and thirty-five patients underwent EUS-FNA for intra-abdominal mass lesions during the study period. One hundred and forty-five patients were excluded due to cystic lesions. Specimens were not available for review in 24 cases; thus, 166 patients were included in the study.” : you should report this part in the result section.

-Line 150: “ROSE is not available”: “ROSE was not available”.

-Lines 159-166: “The immunohistochemical stains…”: you should mention the clones you used.

-Line 178 “(Pap)”: in my opinion you should use either the entire word or the abbreviation.

-Table 2, “Surgical pathology” column: you should check the concordance with the data reported in the text.

-Table 3: you should delete the “%” signs.

-Table 4: you should delete the word “abbreviation”.

-Line 290: where does the data “13 patients” come from?

-Lines 292-293, “Of 86 patients without the diagnosis of malignancy based on cytological evaluation (inadequate = 28, negative for malignancy = 58)”: you should report in the text the results of the FNAB of all the patients in this study.

---

## [Author Response · Author response to Decision Letter 1]

30 Jan 2022

Dear Editor,

We would like to thank the editor and both reviewers for reviewing our manuscript and providing constructive criticisms. As suggested, we have revised the manuscript point-by-point in response to the reviewers’ questions. 

Reviewers' comments:

Reviewer #1: The authors have assessed the issues clarifying doubts and appropriately modifying the paper, therefore in my opinion the article can be submitted for publication.

Response: We are grateful for the reviewer’s input which helps improve the quality of our manuscript. Your time is greatly appreciated.

Reviewer #2: I would like to thank the authors for all the answers to my comments and the revision of the manuscript. In my opinion there are still some issues that need to be addressed before publication:

Response: We thank the reviewer for taking the time to review our manuscript and providing helpful comments. We have revised our manuscript to address the reviewer’s comments, and a point-by-point response to the specific questions is provided. 

-The word "histology" is still often used directly in place of “cell block preparation”. In my opinion, the terminology must be checked through all the text (considering also the abstract and the tables).

Response: Thank you for pointing this out. To make the terminology consistent throughout the article, we have changed the terminology regarding the tissue processing method (cytology, cytological evaluation, histology, cell block preparation) throughout the manuscript. Changes have been made to all sections, including the manuscript title, abstract, introduction, materials and methods, results, discussion, conclusion, tables, and the figure legend. 

1. The manuscript title has been changed to “The diagnostic performance of combined conventional cytology with smears and cell block preparation obtained from endoscopic ultrasound-guided fine needle aspiration for intra-abdominal mass lesions”.

2. The word “cytological analysis” was replaced by “cytology” or “conventional cytology” or “cytologic smears” when appropriate. 

3. The word “histology” was replaced by “cell block preparation” or “cell block preparation for histological assessment” when appropriate.

-Line 31 and others: ”combined cytological”: you should change it to “cytology” or “conventional cytology” through all the text.

Response: Changes have been made as suggested. “combined cytological….” was changed to “combined conventional cytology and cell block preparation…..”

-Lines 127-130: “Three hundred and thirty-five patients underwent EUS-FNA for intra-abdominal mass lesions during the study period. One hundred and forty-five patients were excluded due to cystic lesions. Specimens were not available for review in 24 cases; thus, 166 patients were included in the study.” : you should report this part in the result section.

Response: Changes have been made as suggested. Lines 127-130 were moved to the result section. 

-Line 150: “ROSE is not available”: “ROSE was not available”.

Response: Change has been made as suggested.

-Lines 159-166: “The immunohistochemical stains…”: you should mention the clones you used.

Response: A summary of antibodies and clones used in this study has been put together in Supplementary table 1 (please see the attached supplementary table 1). The following sentence, “The information about antibody clones are provided in Supplementary Table 1” was added to the text under the Materials & Methods section, subheading “Cytological smears and cell block preparation”. 

-Line 178 “(Pap)”: in my opinion you should use either the entire word or the abbreviation.

Response: Pap was changed to Papanicolaou.

-Table 2, “Surgical pathology” column: you should check the concordance with the data reported in the text.

Response: The concordance between the data in table 2 and the text was reviewed. Table 2 has been revised as suggested.

-Table 3: you should delete the “%” signs.

Response: “%” signs were removed. 

-Table 4: you should delete the word “abbreviation”.

Response: The word “abbreviation” below Table 4 was removed. 

-Line 290: where does the data “13 patients” come from?

Response: We intended to state that in cases of surgical pathology proven malignancy, the diagnostic accuracy of EUS-FNA was 100%. The number of patients with surgical pathology was 16, not 13. “13” was a typo. However, we elected to remove this sentence as the number of patients with surgical pathology proven malignancy was quite limited, as mentioned in the study limitations, and added limited value. 

-Lines 292-293, “Of 86 patients without the diagnosis of malignancy based on cytological evaluation (inadequate = 28, negative for malignancy = 58)”: you should report in the text the results of the FNAB of all the patients in this study.

Response: Of 86 patients who had either inadequate cytology or cytological diagnosis of negative for malignancy, 58 patients had adequate specimens for cell block preparation. Of these, 8 patients (9.3%) were found to have malignancy, including pancreatic and metastatic cancer. Other diagnoses included tuberculous lymphadenitis (n = 10), reactive lymph nodes (n = 9), abscess (n = 3), GIST (n = 1), benign pancreatic tissue (n = 1). The diagnoses could not be made by cell block evaluation in 26 patients. Among these, malignancy (n = 16), tuberculosis (n = 6), and other benign conditions (n = 4) were discovered during clinical and imaging follow-up. These results have been added as suggested.

---

## [Editor Report · Decision Letter 2]

2 Feb 2022

The diagnostic performance of combined conventional cytology with smears and cell block preparation obtained from endoscopic ultrasound-guided fine needle aspiration for intra-abdominal mass lesions

PONE-D-21-31814R2

Dear Dr. Pausawasdi,

We’re pleased to inform you that your manuscript has been judged scientifically suitable for publication and will be formally accepted for publication once it meets all outstanding technical requirements.

Kind regards,

Vincenzo L'Imperio

Academic Editor

PLOS ONE

---

## [Editor Report · Acceptance letter]

14 Mar 2022

PONE-D-21-31814R2 

The diagnostic performance of combined conventional cytology with smears and cell block preparation obtained from endoscopic ultrasound-guided fine needle aspiration for intra-abdominal mass lesions 

Dear Dr. Pausawasdi:

I'm pleased to inform you that your manuscript has been deemed suitable for publication in PLOS ONE. Congratulations! Your manuscript is now with our production department. 

Kind regards, 

on behalf of

Dr. Vincenzo L'Imperio 

Academic Editor

PLOS ONE